# Quantitative Image Quality Metrics of the Low-Dose 2D/3D Slot Scanner Compared to Two Conventional Digital Radiography X-ray Imaging Systems

**DOI:** 10.3390/diagnostics11091699

**Published:** 2021-09-17

**Authors:** Ahmed Jibril Abdi, Bo R. Mussmann, Alistair Mackenzie, Oke Gerke, Benedikte Klaerke, Poul Erik Andersen

**Affiliations:** 1Department of Clinical Research, University of Southern Denmark, 5000 Odense C, Denmark; bo.mussmann@rsyd.dk (B.R.M.); oke.gerke@rsyd.dk (O.G.); peandersen@health.sdu.dk (P.E.A.); 2Region of Southern Denmark, Clinical Engineering Department, Area of Diagnostic Radiology, 5000 Odense C, Denmark; benedikte.klarke@rsyd.dk; 3Department of Radiology, Odense University Hospital, 5000 Odense C, Denmark; 4National Coordinating Centre for the Physics of Mammography, Royal Surrey NHS, Foundation Trust, Guildford GU2 7XX, UK; alistairmackenzie@nhs.net; 5Department of Nuclear Medicine, Odense University Hospital, 5000 Odense C, Denmark

**Keywords:** chest X-ray, knee X-ray, quantitative image quality metrics

## Abstract

The aim of this study was to determine the quantitative image quality metrics of the low-dose 2D/3D EOS slot scanner X-ray imaging system (LDSS) compared with conventional digital radiography (DR) X-ray imaging systems. The effective detective quantum efficiency (eDQE) and effective noise quantum equivalent (eNEQ) were measured using chest and knee protocols. Methods: A Nationwide Evaluation of X-ray Trends (NEXT) of a chest adult phantom and a PolyMethylmethacrylate (PMMA) phantom were used for the chest and knee protocols, respectively. Quantitative image quality metrics, including effective normalised noise power spectrum (eNNPS), effective modulation transfer function (eMTF), eDQE and eNEQ of the LDSS and DR imaging systems were assessed and compared. Results: In the chest acquisition, the LDSS imaging system achieved significantly higher eNEQ and eDQE than the DR imaging systems at lower and higher spatial frequencies (0.001 ≤ *p* ≤ 0.044). For the knee acquisition, the LDSS imaging system also achieved significantly higher eNEQ and eDQE than the DR imaging systems at lower and higher spatial frequencies (0.001 ≤ *p* ≤ 0.002). However, there was no significant difference in eNEQ and eDQE between DR systems 1 and 2 at lower and higher spatial frequencies (0.10 < *p* < 1.00) for either chest or knee protocols. Conclusion: The LDSS imaging system performed well compared to the DR systems. Thus, we have demonstrated that the LDSS imaging system has the potential to be used for clinical diagnostic purposes.

## 1. Introduction

The low-dose 2D/3D EOS slot scanner (LDSS) imaging system is a relatively new imaging modality that emits less radiation to patients when compared with conventional digital radiography (DR) imaging systems [1]. The LDSS and DR imaging systems are both based on X-ray technology. However, the two modalities differ, mainly with regard to image acquisition. The LDSS imaging system acquires X-ray images using slot-scan technology. The beam profile of the LDSS geometry is a fan-shaped beam across the collimator plane (parallel to the floor). The detector and the X-ray tube follow each other synchronously and scan the object (patient) from top to bottom. In conventional DR systems, the X-ray beam is cone-shaped and the dose follows the law of being reduced by the square root of the focus to detector distance (inverse square law). Another difference between the LDSS imaging and conventional DR systems is the detector technology. LDSS has a gaseous radiation detector and a micro-grid ionisation chamber that is more sensitive than the conventional solid-state detectors used in DR systems [2,3]. In the LDSS imaging system, the detector array extends along a plane that uses two collimator slots called the detector and object collimator slots. The collimated X-ray beam has a width of 0.5 mm. This small coverage area of the image detector results in a significant reduction in scattered radiation reaching the detector. The X-ray tube and X-ray generator for the LDSS imaging system are the same as those used in computed tomography (CT). LDSS is a bi-plane with two X-ray tubes and two image detectors that move synchronously in lateral and frontal projections. The 2D images of the LDSS imaging system make it possible to obtain 3D images for angulation and distance measurements [4]. In this study, only one plane is acquired.

LDSS imaging technology is mostly used for diagnostic purposes in patients with scoliosis or leg length discrepancy [5,6]. The imaging system has not been optimised to acquire image quality comparable to that of the conventional DR imaging system and thus, it is not typically used as a diagnostic tool in general radiology. LDSS has been described in more detail in other studies [2,7,8]. There is value in piloting this equipment because it renders a lower radiation dose to patients compared to a DR system [9]. Thus, it is of interest to test and characterise the LDSS system to determine whether it can produce diagnostic clinical images for other types of examinations.

Detective quantum efficiency (DQE) determines the ability of a detector to transfer the signal-to-noise ratio (SNR) within the detector as a function of spatial frequency [10]. The DQE helps characterise digital image detectors and has been used to compare detectors [11,12]. However, DQE is not optimal for assessing overall image quality and performance of the entire imaging system because it does not include impact from the magnification, focal spot blurring, the appearance of scatter radiation generated by the patient, or the presence of anti-scatter grids [13]. Samei et al. [14] have shown that slot scanner systems can have relatively low DQE compared to flat panel detectors, but still produce clinically acceptable images. Alternative metrics to quantitatively characterising the whole imaging system have been developed [15,16], and these can provide a more useful comparison of imaging systems, particularly for scanning systems such as the LDSS imaging system. In this study, the effective DQE (eDQE) metric is used as an objective and quantitative approach [17].

The eDQE quantitative metric takes into account magnification, focal spot blurring, the scattered radiation caused by the patient, and the presence of anti-scatter grids of the imaging system [13,17,18,19]. The eDQE is computed in a similar manner as DQE, but eDQE formulation requires the calculation of the scatter fraction, the transmission fraction of the phantom, and the magnification at the plane of interest [20].

The eDQE is a measure of the system’s efficiency and is thus useful for comparing imaging systems. Measurements of eDQE have been undertaken on a number of imaging systems for chest protocols but never for knee protocols [14,21,22,23]. However, a high eDQE does not necessarily translate into acceptable image quality and depends on the dose level used. The image quality metric of effective noise equivalent quanta (eNEQ) is considered to be the effective number of X-ray quanta contributing to the image [12]. The eNEQ correlates better with image quality. Thus, the higher the eNEQ, the better the clinical image quality [23,24]. The LDSS imaging system is designed to render low-dose imaging. Systems using scanning imaging techniques have been successfully applied in chest and breast imaging [14,25,26], and the suitability of the LDSS imaging system is considered in the present work for acquiring chest and extremity radiographic examinations. Chest radiography is a cheap, fast and well-established procedure that remains the mainstay for diagnosing many thoracic diseases [27]. Thorax anatomical regions contain several different tissue contrasts, including low and high contrast resolution. The radiographic procedure is also commonly used to acquire images of the extremities including knee acquisition. The knee contains bone regions with high contrast resolution and joints. These two clinical protocols can be used to optimise the low and high contrast sensitivity of the imaging systems. Therefore, in this study we chose to use chest and knee protocols to assess quantitative image quality. Previously, we compared the radiation dose and image quality measured by using a contrast detail test object [1]. The results confirmed that radiation doses were lower for the LDSS system than for two conventional DR systems but that both modalities yielded equally adequate image quality.

The eDQE was initially developed to measure image quality in chest examination protocols, but its use was later extended to other anatomical areas, including breast examinations [13,17,20,28]. In the present study, the technique was also extended to the knee examination protocol. The aim of this study was to measure the eDQE and eNEQ of chest and knee examinations using the LDSS imaging system compared to two conventional DR imaging systems.

## 2. Materials and Methods

The following three imaging systems were used to determine the quantitative image quality metrics:

LDSS: The LDSS imaging system (EOS SA, Paris, France) allows the acquisition of two simultaneous X-ray images, which are the posterior-anterior and lateral projections. From these two-dimensional (2D) images, it is possible to derive three-dimensional (3D) images, and it is possible to activate only one source of the system.

DR system 1: The Philips Digital Diagnost (DiDi) DR X-ray imaging system (Philips Healthcare, Best, The Netherlands) is a conventional wall stand DR X-ray imaging system using a Trixell flat panel detector (with detector technology of Cesium Iodide (CsI), Pixel size 143 µm).

DR system 2: The Siemens Ysio DR X-ray imaging system (Siemens Healthineers GmbH, Forchheim, Germany) is a conventional wall stand DR system using a Trixell flat panel receptor (with detector technology of CsI, pixel size 139 µm).

A Piranha 657 solid-state dosimeter (RTI Group, Mölndal, Sweden) was used to measure incident air kerma (IAK) and detector air kerma (DAK) for all imaging systems.

Quantitative parameter calculations, including effective modulation transfer function (eMTF) and effective normalised noise power spectrum (eNNPS), were performed using open source IQWorks 0.7.2 quantitative software analysis for CT, mammography and digital radiography from SourceForge open source software, Slashdot Media, San Diego, CA, USA (https://sourceforge.net/projects/iqworks/ (accessed on 16 September 2021)).

SpekCalc software programme (The Institute of Cancer Research, London, UK), was used to calculate the number of X-ray photons in a section of beam contributing to a given air kerma (AK) [29,30,31], (http://spekcalc.weebly.com/ (accessed on 16 September 2021)).

To assess and calculate effective quantitative metrics (eDQE, eMTF and eNNPS, eNEQ), an in-house phantom designed by the American Food and Drug Administration was used [13,18]. The phantom was primarily designed for the nationwide evaluation of the X-ray trend (NEXT) programme. The NEXT phantom was designed to provide the same exposure as an adult chest radiography. The phantom comprises various attenuation materials such as Polymethyl methacrylate (PMMA) attenuation plates and aluminium foils. The NEXT Phantom measures 30 cm × 30 cm and contains attenuation materials, including PMMA blocks, aluminium sheets and an air gap. The air gap in the phantom simulates the thoracic cavity. The NEXT phantom was used as a patient-equivalent phantom in the chest protocol for all imaging systems (Figure 1).

30 cm × 30 cm × 15 cm of PMMA blocks were used to determine the quantitative image quality metrics of the knee examination protocol.

A high-contrast object of 1 mm thick tungsten sheet with a purity of 99.95% (Advent research materials Ltd, Eynsham Oxford, UK, OX29 4JA), which meets the specification set out in IEC 62220-1 [32,33,34,35,36], was used to determine the edge eMTF measurement for all imaging systems.

An in-house beam-stopping device phantom containing an 11 × 11 array of lead cylinders embedded in 10 mm thick PMMA was used to assess the scatter fraction of all imaging systems (Figure 2).

### 2.1. Examination Protocols and Configurations

Chest and knee examination protocols were used to evaluate the quantitative image quality metrics. These two protocols were also used in a previous study of the imaging systems [1]. The clinical examination settings for all imaging systems used in the present study are shown in Table 1. The default exposure parameter settings of DR imaging systems are the same as those generally used in clinical practice. However, the clinical examination settings for the LDSS imaging system for the chest and knee protocols were optimised based on a previously conducted study [1].

The protocol configurations including source image distance (SID), tube voltage (kV), additional filtration and other relevant parameters in the chest and knee protocols are shown in Table 2. An anti-scatter grid was used on the chest protocol of both DR systems, whereas all LDSS images and knee images in the DR imaging systems were acquired without an anti-scatter grid.

Speeds 6 and 8 are the optimised LDSS scan speeds reported in a previous study for chest and knee protocols, respectively [1]. The scan speed is a steady linear speed at which the objects/patients are scanned. A low linear scan speed increases dose in an inverse proportion.

### 2.2. Determination of Quantitative Parameters

The following parameters were measured as part of calculating the eDQE and eNEQ.

#### 2.2.1. Detector Response

The relationship between pixel values and AK (detector response function) was measured for each imaging system and radiographic setting for the chest and knee protocols. These measurements were then used to linearize DICOM images before the images underwent qualitative analysis. To measure the detector response function, the phantom was placed between the source and the image detector close to the X-ray tube. Images were acquired using a range of 10 different dose levels (tube loads). All image data were obtained using the minimal processing algorithm from the imaging system. The AK of each image acquisition was measured. AK was measured free-in-air using a calibrated solid-state dosimeter with the same exposure parameters used in the image acquisitions. The dosimeter was placed close to the image detector plane. Mean pixel values for the images were measured. The detector response function was a fit of mean pixel values against AK incident to the detector.

#### 2.2.2. Effective DQE

eDQE was calculated for all imaging systems and settings for chest and knee protocols using the following Equation [13]
(1)$$\mathrm{eDQE}\;\left(f\right)=\frac{\mathrm{eMTF}{\left(f\right)}^{2}\cdot {\left(1-\mathrm{SF}\right)}^{2}}{\mathrm{eNNPS}\left(f\right)\cdot \mathrm{TF}\cdot \mathrm{AK}\cdot q}$$
where f is the spatial frequency corrected to object plane (mm^−1^), eMTF is the calculated effective modulation transfer function as a function of the spatial frequency at the object plane, SF is the scatter fraction, and eNNPS(f) is the normalised noise power spectrum as a function of the spatial frequency at the object plane. TF is the transmission factor through the phantom, AK (µGy) is the AK measured without the phantom being corrected to the detector plane, and *q* (µGy^−^^1^·mm^−^^2^) is the estimated number of X-ray photons per unit area per AK.

#### 2.2.3. Effective Noise Equivalent Quanta (eNEQ)

eNEQ (mm^−2^) is a measure of the overall image quality of the imaging system, which is considered equal to the number of X-ray quanta equivalents in the X-ray image [37]. eNEQ of all imaging systems and examination protocols were calculated using the following equation introduced by Ertan et al. [12]:(2)$$\mathrm{eNEQ}\left(f\right)=\frac{\mathrm{eMTF}{\left(f\right)}^{2}\cdot {\left(1-\mathrm{SF}\right)}^{2}}{\mathrm{eNNPS}\left(f\right)}\;$$

#### 2.2.4. Determination of the Transmission Factor (TF) and AK

The AK of scatter-free geometric settings for all imaging systems and clinical procedures was measured using an RTI solid-state dosimeter, where the phantom was placed between the X-ray source and dosimeter. To minimise the contribution of the scatter radiation, the phantom was placed as far as possible from the dosimeter, which was approximately 40–100 cm. The AK for each configuration was measured with and without a phantom using the narrow beam configuration. TF was calculated using the ratio of AK measured with and without the phantom. The TF for all the imaging systems and the settings was determined using the following Equation (3):(3)$$\mathrm{TF}=\frac{{\mathrm{AK}}_{+}}{{\mathrm{AK}}_{-}}$$
AK_+_ and AK_−_ are AK measured with and without phantom, respectively.

The incident air kerma (IAK) at the detector plane for all imaging systems and the setting obtained by multiplying AK is measured by the ratio of the source-to-dosimeter distance (SDD) to the SID. The formula for calculating IAK in LDSS imaging systems is:(4)$$\mathrm{IAK}={\mathrm{AK}}_{+}\cdot \frac{\mathrm{SDD}}{\mathrm{SID}}$$
where SDD is a source-to-dosimeter distance, and SID is source-to-image detector. To calculate the incident k of the DR X-ray imaging systems, the inverse square law formula was used:(5)$$\mathrm{IAK}={\mathrm{AK}}_{+}\cdot \frac{{\mathrm{SDD}}^{2}}{{\mathrm{SID}}^{2}}$$

AK at the detector plane (DAK) was calculated using the following Equation (6).
(6)DAK=IAK·TF

#### 2.2.5. Effective Modulation Transfer Function (eMTF)

eMTF of all imaging systems and clinical protocol configurations were determined using a test object with a polished edge. To determine the eMTF of all imaging systems and examination protocol configurations, a 1 mm tungsten plate was used to produce a high-contrast image of the edge. The tungsten edge device was placed 5 cm in front of the entrance to the patient-equivalent phantom (Figure 3). The edge test object was centred with a 3° tilt angle in the vertical direction. Images of the edge were acquired at 5 times the standard exposure level. The edge was shifted slightly between each exposure.

To determine eMTF, a processed and linearized pixel value from an 80 mm × 80 mm ROI was projected along the edge angle with 0.1 subdivisions of the pixel pitch. This created a super-sampled ESF. The ESF was then differentiated to become the line spread function (LSF). The eMTF was finally calculated by applying the modulus of the Fourier transform of LSF. eMTF was normalised so that eMTF would be 1 at the zero frequency and the eMTF was averaged by 5 acquired edge images.

#### 2.2.6. Effective Normalised Noise Power Spectrum (eNNPS)

eNNPS is an important parameter for quantifying the noise content of an X-ray image and describes the image noise as a function of spatial frequency. eNNPS was measured using a NEXT phantom and a PMMA attenuation plate (NEXT phantom and 13 cm thick PMMA for chest and knee protocols, respectively). The patient-equivalent phantom was placed in the middle of the X-ray beam, in the standard patient position (Figure 3). Four images of the homogeneous PMMA phantom were then acquired. The images were then uploaded to the eNNPS calculating software algorithm. The software algorithm selects an ROI pixel size area of 1024 × 1024 at the centre of the homogenous phantom images. Next, ROI was divided into square sub-regions with a pixel size of 256 × 256 and with an overlap of 128 pixels in both the horizontal and vertical planes. The 2D noise power spectrum was calculated with the squared modulus of the Fourier transform and was averaged for all sub-regions. The 2D eNPS is normalised by the squared linearised pixel value to obtain 2D eNNPS. One-dimensional eNNPS was calculated averaging 2D eNNPS in the vertical direction excluding on-axis data.

#### 2.2.7. Scatter Fraction (SF)

SF was obtained using a beam-stop device technique [38,39]. The beam-stop device was placed at the radiation entrance surface of the patient-equivalent phantom, and images that contained both the beam-stopping cylinders and the phantom were acquired.

The attenuated signal (MPV_Lead_cylinders_) was the average signal measured behind the central 25 lead cylinders in the central portion of the radiographic image. The SF of all of the imaging systems and configurations was then calculated from the ratio of attenuated signal to the average background signal (MPV_Background_) without lead [19]. The following Equation (7) was used to calculate SF:(7)$$\mathrm{SF}=\frac{{\mathrm{MPV}}_{\mathrm{Lead}\_\mathrm{cylinders}}}{{\mathrm{MPV}}_{\mathrm{Background}\;}}$$

#### 2.2.8. Estimation of *q*-Value

*q*-value is a factor that determines the estimation of eDQE, which corresponds to the ideal squared SNR [40]. *q*-value was obtained using the SpekCalc X-ray spectra computing software package [29,30,31].

The SpekCalc software input parameters were adjusted and customised to the output parameters of the imaging systems. Next, the X-ray energy spectra and an AK rate were computed for each imaging system configuration of tube voltage [40]. The photon fluence, Φ, is defined as the quotient of *d*N by *d*A [41], as shown in Equation (8):(8)$$\Phi =\frac{dN}{dA}\;({\mathrm{mm}}^{-2}),$$
where *d*N is the number of photons incident on the cross-sectional area *d*A of the spectrum.
(9)$$q=\left(\frac{\Phi_{\mathrm{tot}}\;}{\mathrm{AK}}\right)=\left({\mathrm{mm}}^{-2}\cdot {\;µ\mathrm{Gy}}^{-1}\right)$$
where (Φ_tot_) is the total photon fluence (Φ_tot_) per mAs calculated by SpecCalc software. AK is the AK per mAs for the distribution Φe (E) [41].

#### 2.2.9. Effective Quantitative Metrics Measurement Setup

The measurement setup of the effective quantitative measurements including eMTF, eNNPS, and scatter fraction is illustrated in Figure 3. Both the MTF edge device and beam- stop device were removed from the radiation beam field during the acquisition of eNNPS images, which were acquired with only the patient-equivalent phantom placed at the entrance of the image detector. The beam-stop device was also removed during eMTF and eNNPS acquisitions for both chest and knee protocols. The acquisition setup diagram is shown in Figure 3. All exposures in the LDSS imaging system on both the chest and knee protocols were obtained without an anti-scatter grid. In the knee examination acquisitions, the NEXT phantom was replaced by 15 cm PMMA blocks.

### 2.3. Statistical Analysis

In order to compare eDQE values between the three imaging systems, Kruskal–Wallis test was used across all groups [42]. Given the statistically significant differences according to the global hypothesis testing using the Kruskal–Wallis test, the eDQE values derived by the three systems were then compared pairwise using Wilcoxon–Mann–Whitney test in the sense of a hierarchical testing procedure. These methods are rank-based nonparametric statistical tests which do not depend on distributional assumptions and which prove to be especially useful for small sample sizes [43,44,45]. Linearity of detector response was assessed using simple linear regression. The level of significance was 5% without adjustment for exploratory, multiple testing. Statistical Package for the Social Sciences (SPSS), Release 26.0.0.0, New York, NY, USA, was applied for all statistical analyses.

## 3. Results

### 3.1. Detector Response (Linearity)

The detector response functions of the DR and LDSS X-ray imaging systems were shown to be linear (Figure 4). The linear detector response function of the imaging systems was determined by plotting the mean pixel value against detector AK.

The determination coefficients from linear regression (R^2^) of all imaging systems in both chest and knee examination configurations range between 0.98 and 0.99. However, the detector response of the LDSS imaging system was assessed using a five-order polynomial, which takes into account the non-linear response of the LDSS imaging systems. Therefore, the detector response data of LDSS imaging system is linearized through a slight manipulation of the data [7]. The detector response functions of X-ray imaging systems for the chest and knee protocols are shown in Figure 4.

The results of all assessed parameters in relation to effective DQE (eDQE) and effective NEQ (eNEQ) calculations are summarised in Table 3. The LDSS imaging system has higher maximum peak values for both eDQE and eNEQ in both the chest and knee protocols compared to the DR imaging systems. The SF is lower in the LDSS than in the DR systems in both knee and chest protocols, which can be explained by the higher maximum peak of eNEQ and eDQE for the LDSS imaging system.

### 3.2. Effective Modulation Transfer Function

For the DR imaging systems, there are small variations of eMTF in both the chest and knee protocols. DR system 1 has a marginally higher eMTF at a higher frequency compared to DR system 2. However, most variation extends between eMTF obtained in the LDSS imaging system and DR imaging systems. The pre-sampled eMTF of the LDSS imaging system drops at higher frequencies for both the chest and knee acquisitions faster than the DR imaging systems, which may be explained by the higher pixel size of the LDSS detector compared with the DR system detectors. In addition, the scan movement of the LDSS imaging system may introduce blur and reduces the eMTF in that direction.

The calculated pre-sampled eMTF in the vertical plane for all imaging systems and examination configurations is presented in Figure 5 as a function of spatial frequency.

### 3.3. Effective Normalised Noise Power Spectrum

The results of eNNPS of the vertical plane for all imaging systems in the chest and knee protocols are shown in Figure 6. According to the comparison of eNNPS for both the chest and knee protocols, eNNPS of DR systems is higher than that of the LDSS imaging systems. This may be explained by higher noise elements in the DR imaging systems due to fewer X-ray photons (primary and scattered) reaching the detector of the DR systems compared to LDSS imaging system.

### 3.4. Effective Detective Quantum Efficiency (eDQE)

eDQE of all imaging systems and examination configurations was determined using the parameters in Table 4 and Equation (1).

As shown in Figure 7, the eDQE results obtained in all imaging systems are plotted in the same coordinate systems as a function of spatial frequency. Moreover, maximum values of eDQE for all imaging systems and protocols are summarised in Table 4.

According to the statistical comparison of eDQE values across the imaging systems, LDSS achieved significantly higher eDQE values for both lower and higher frequencies than both DR imaging systems in both chest and knee acquisitions.

In the chest protocol, LDSS achieved significantly higher eDQE than DR systems 1 and 2 at lower (*p* < 0.001) and higher (0.03 ≥ *p* ≤ 0.04) frequencies. However, no significant difference was found in the eDQE between the DR systems at the lower (*p* = 0.100) and higher (*p* = 0.85) frequencies in the chest acquisition.

For the knee protocol, the LDSS imaging systems also achieved significantly better eDQE than the DR systems, both at lower (*p* < 0.001) and higher (*p* < 0.001) frequencies. However, no significant difference in eDQE was found between the DR systems at lower (*p* = 0.44) and higher (*p* = 0.99) frequencies for the knee acquisition.

### 3.5. Effective Noise Equivalent Quanta (eNEQ)

For the chest acquisition, the LDSS imaging system achieved higher eNEQ values than the DR imaging systems at both higher and lower frequencies.

In the chest acquisition, statistical comparison results showed that LDSS has higher eNEQ values at the low (*p* < 0.001) and higher (*p* = 0.022) frequencies than in the DR system 1. The LDSS imaging system also achieved a higher eNEQ than DR system 2, both at low (*p* < 0.001) and higher (*p* = 0.034) frequencies. However, there were no significant differences in the eNEQ values obtained at the lower (*p* = 0.76) and higher (*p* = 0.80) frequencies between DR imaging system 1 and 2 for the chest acquisition.

For the knee acquisition, the LDSS imaging systems also achieved higher eNEQ values than the DR imaging systems at both higher and lower frequencies.

The LDSS imaging system achieved a significantly higher eNEQ value in the lower frequencies than DR system 1 and DR system 2 (*p* < 0.001) and (*p* < 0.001), respectively. The LDSS imaging system also achieved significantly higher eNEQ values at higher frequencies than DR system 1 and DR system 2 (*p* = 0.002) and (*p* = 0.002), respectively.

There was no significant difference between eNEQ values obtained from DR system 1 and DR system 2 for either lower (*p* = 0.59) or higher (*p* = 0.89) frequencies in the knee acquisition. The eNEQ comparison results obtained for all imaging systems for both chest and knee protocols are shown in Figure 8.

### 3.6. Statistical Comparison

The eDQE graphical comparison introduced in Figure 7 shows that the order of the imaging systems is dependent on spatial frequencies. Therefore, the spatial frequencies were split into lower and higher frequencies to conduct stratified analyses.

The Kruskal–Wallis statistical comparison of eDQE values presented in Table 4 shows that statistically significant differences across the imaging systems were observed at lower and higher spatial frequencies for both the chest and knee acquisitions.

Since Kruskal–Wallis comparisons between eDQE values for all three systems and examination protocols indicated significant differences in the above subgroup analyses, pairwise comparisons for all three systems and examination protocols were supplemented. Mann–Whitney U pairwise comparisons of eDQE values for all imaging systems and acquisitions are shown in Table 5.

Box-plots of eDQE values for low and high frequencies in knee and chest protocols are shown in Figure 9. The LDSS imaging system has a higher magnitude of eDQE than the DR imaging systems at lower and higher spatial frequencies for chest and knee protocols.

The LDSS imaging system has a higher eNEQ magnitude at lower frequencies than the DR imaging system in the chest protocol and higher frequencies in the knee protocol; therefore, stratified analyses were performed by splitting eNEQ values into low and high spatial frequencies. The spatial frequencies of eNEQ for chest and knee protocols are split into 0.00–1.25 mm^−1^ and 1.35–2.55 mm^−1^. Kruskal–Wallis test comparison of eNEQ values for both examination protocols is shown in Table 6.

As shown in Table 7, eNEQ intersystem differences were observed at the lower and higher spatial frequencies of the chest and knee protocols. Pairwise comparisons of the eNEQ values across the imaging systems in chest and knee acquisitions at lower and higher spatial frequencies are shown in Table 7.

The box-plots of eNEQ for all imaging systems in the chest and knee protocols are shown in Figure 10. The LDSS imaging system has a higher magnitude of eNEQ than the DR imaging systems at both lower and higher frequencies for the chest and knee protocols.

## 4. Discussion

Quantitative image quality metrics, including eMTF, eNNPS, eDQE and eNEQ, were obtained in an LDSS imaging system and in two DR imaging systems for chest and knee examination protocols. The DR imaging systems performed slightly better in some of the assessed parameters, including eMTF, particularly at the higher spatial frequencies. Although LDSS imaging has a lower performance of eMTF, especially at the higher frequencies, the system has a better noise property than the DR imaging systems, resulting in a higher eNEQ and eDQE.

According to the image quality parameter eNEQ and eDQE results presented in Table 3 and the subsequent statistical analysis, the LDSS imaging system has shown better eDQE and eNEQ characteristics compared to the DR imaging systems in both chest and knee examination protocols. The higher maximum peaks of eDQE and eNEQ in the LDSS imaging system most likely occurred because the LDSS imaging system had lower *q*-values, SF and eNNPS than the DR imaging systems in the chest and knee protocols. The higher tube voltage used in the DR imaging systems will result in a larger *q*-value than in the LDSS imaging systems.

The SF factors for images from the LDSS imaging system are lower than the SF for images from the DR imaging systems for both knee and chest protocol acquisitions because LDSS is a scanning system with intrinsic scatter rejection. In the knee protocol, the LDSS imaging system achieved lower TF, SF and eNNPS values than did the DR imaging systems, resulting in higher eDQE and eNEQ peaks.

The main benefit of the LDSS imaging system is that it exposes the patient to a lower radiation dose compared to conventional DR systems [1]. Therefore, an LDSS imaging system with optimised image quality could be suitable for diagnosing radiation-sensitive patients, including children and adolescents. However, the LDSS imaging system requires a much longer exposure time than the DR system, and it is uncomfortable for patients to hold their breath for longer periods. This long exposure time can also cause motion artefacts in the patient’s internal and external organs.

The present study was based on a quantitative phantom measurement study. It did not include a clinical image quality assessment of the systems, which may be a limitation. However, the strength of this study is that the image quality of the systems was objectively assessed for both the chest and knee protocols and did not depend on a subjective assessment by an observer. The LDSS imaging system has been optimised to provide image quality comparable to that of conventional DR chest and knee radiography. The scan speed for the knee protocol was increased from speed 6 to speed 8 from the standard settings used. Similarly, the scan speed for the chest protocol was increased from speed 4 to speed 6. Increasing scanner speeds resulted in increased radiation doses of 36% and 50% for knee and chest protocols, respectively, compared to the default setup dose level [1]. Due to the extensive optimisation of the LDSS imaging system and the use of a different phantom in the current study, the assessed quantitative image quality of the LDSS imaging system in the present study is not directly comparable with that of a previously performed similar study [40]. In contrast to the LDSS imaging systems, the DR radiographic imaging systems used in the current study were not optimised, particularly for the thoracic protocol acquisitions. Chest acquisitions for the DR imaging systems operate at a higher tube voltage (kV) than the commonly used clinical chest configurations. This higher tube energy operating setup of the DR imaging systems is based on current, clinically applicable standard chest protocol configurations in the radiology department where the present study was performed.

Therefore, the clinical chest configurations of the DR imaging systems have not been optimised. However, we compensated for the high beam energy of the chest DR imaging system acquisitions by having a larger SID (300 cm). Despite the difference between the setup of the imaging systems used in the current study and previous studies, there are still some comparable areas between the parameters measured in the current study and the earlier [40] studies.

The quantitative metrics assessed in this study for the LDSS X-ray imaging system differ from those previously obtained by other researchers [40]; in the present study the acquisition parameters were optimised and different phantoms were used. Although the dose levels, phantom sizes and acquisition parameter settings in the previous study are not equal to the entire acquisition parameter setup and phantom sizes for the current study, some tube voltage level settings can be compared. In the previous study, a maximum peak eDQE value of 0.091 was obtained at a tube voltage of 70 kV. The corresponding maximum peak eDQE value obtained in the current study was 0.11 at a tube voltage of 68 kV.

In another study, eDQE was evaluated at different settings of DR imaging systems [20]. In this work, the exposure setting, examination protocol, tube voltages and phantoms were different from those used in the present study [20]. Much lower eNNPS values were assessed in that previous study, which resulted in higher eDQE.

Another study evaluated the eDQE metrics of different exposure settings and ranges of tube voltages using ThoraScan slot scanning and DR imaging systems for the chest protocol and the same phantom as used in the current study [20]. However, direct comparisons cannot be made between the results for the respective imaging system and the current study results. The DR imaging system acquisition settings and exposure levels of the previous work resulted in large distributions and a wide range of peak eDQE values (0.025–0.15) at a tube voltage of 81–122 kV. The eDQE obtained in DR imaging systems in the current study is lower than the eDQE in some of the DR imaging systems in this previous study. The eDQE results of DR imaging in the current study range between 0.04–0.05 with a tube voltage of 141–145, which is comparable to the results of some of the DR imaging systems in the previous study. The eDQE metrics for the slot scanner (ThoraScan) imaging system in the previous study were in the range of 0.056–0.15 at a fixed tube voltage (140 kV) and different exposure levels.

The eDQE results of LDSS imaging in the current study were in the range of 0.10–0.11 at a tube voltage of 68–90 kV. Although the clinical setup of the imaging systems for these two studies differ with regard to exposure parameters, exposure levels and SID, both studies showed the same trends in quantitative image quality metrics for eMTF, eNNPS, and eDQE. In this previously conducted study, the DR imaging systems achieved a higher noise and scatter fraction than did the ThoraScan imaging system. Similarly, in the current study, the DR imaging systems have shown higher noise and scatter fraction when compared to the LDSS imaging system.

Another study on DR systems using the same chest phantom as used in the current study assessed the eDQE values for different exposure levels and beam qualities [13]. The achieved SF and eDQE values in this study were in the range of 0.29–0.34 and 0.045–0.12, respectively at a tube voltage range of 90–120 kV. However, SF and eDQE values in the current study obtained at a tube voltage of 141–145 kV were 0.18–0.19 and 0.045–0.051, respectively.

The eNEQ values obtained in the previous study for computed radiography (CR) chest acquisition were approximately 1500 to 2000 with a tube voltage range of 70–120 kV, compared to the current study in which the eNEQ values were in the range of 3240 to 3400 with a tube voltage range of 141–145 kV [12]. Due to the different imaging system configurations, the eNEQ values of these two studies are not directly comparable. Nevertheless, there is a good correlation between the eNEQ assessed in these two studies. In the same study, the determined corresponding eDQE values were 0.025–0.04 for a tube voltage range of 70–120 kV. Similarly, in the present study, eDQE values of 0.045–0.051 were determined for a tube voltage range of 141–145 kV.

The quantitative image quality results of the LDSS imaging system obtained in the present study, complemented by the contrast detail resolution obtained in our previous research, demonstrate that the LDSS imaging system has diagnostic potential in regions other than the spine and lower extremities [1]. Moreover, the overall image quality results obtained in this study indicate that the LDSS imaging system has the potential for more extensive use in clinical diagnostic examinations. However, further clinically based studies are needed to evaluate image quality, as these previous studies are based on physical and technical evaluation of the image quality of the systems and do not take into account clinical challenges such as patient positioning and movement.

This study evaluated the technical and physical image quality of the LDSS imaging system, and our previous study evaluated contrast detail resolution compared to the same DR imaging systems that were compared to the LDSS imaging system in the current study [1]. Further observer investigation of clinical image quality is warranted to definitively determine whether the LDSS imaging system can be used for diagnostic purposes in various clinical examination protocols.

## 5. Conclusions

The overall quantitative image quality metrics obtained in this study show that the LDSS imaging system has better quantitative image quality than DR imaging systems for both chest and knee protocols. Thus, the LDSS imaging system has the potential to produce radiographic images with diagnostic information similar to that of conventional DR imaging systems for chest and knee protocols.

However, as this work was based on a quantitative image quality assessment of the systems, an additional clinically based image quality evaluation is called for to determine the usability of the LDSS imaging system for diagnostic purposes.

## Figures and Tables

**Figure 1 diagnostics-11-01699-f001:**
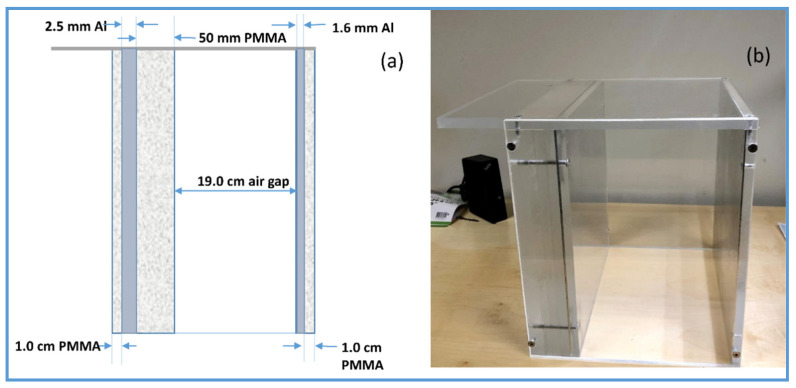
Schematic lateral view of the NEXT patient-equivalent phantom (**a**) and photographic image of the NEXT patient-equivalent phantom for chest protocols (**b**).

**Figure 2 diagnostics-11-01699-f002:**
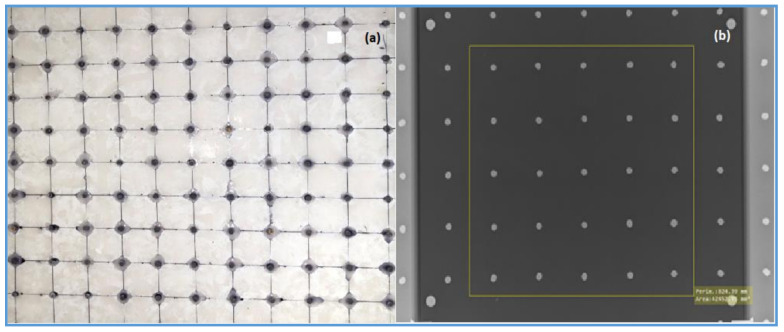
(**a**) Photograph of the in-house beam-stop device (**b**) a radiographic image of beam-stop device with patient- equivalent NEXT phantom with marked central area, used to calculate the scatter fraction factor.

**Figure 3 diagnostics-11-01699-f003:**
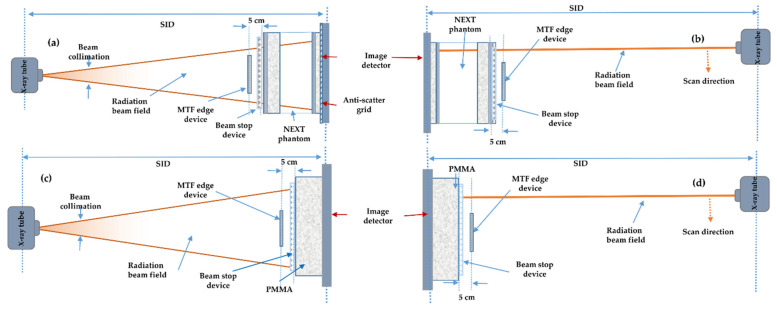
A lateral view of the measurement setup of the quantitative parameters: (**a**) chest protocol measurement setup for the DR imaging systems, (**b**) chest protocol measurement setup for the LDSS imaging system, (**c**) knee protocol measurement setup for the DR imaging systems and (**d**) knee protocol measurement setup for the LDSS imaging system. eNNPS for both protocols was measured using the same measurement setting with only the patient-equivalent phantom and PMMA placed at the entrance of the image detector, SID = source to image distance.

**Figure 4 diagnostics-11-01699-f004:**
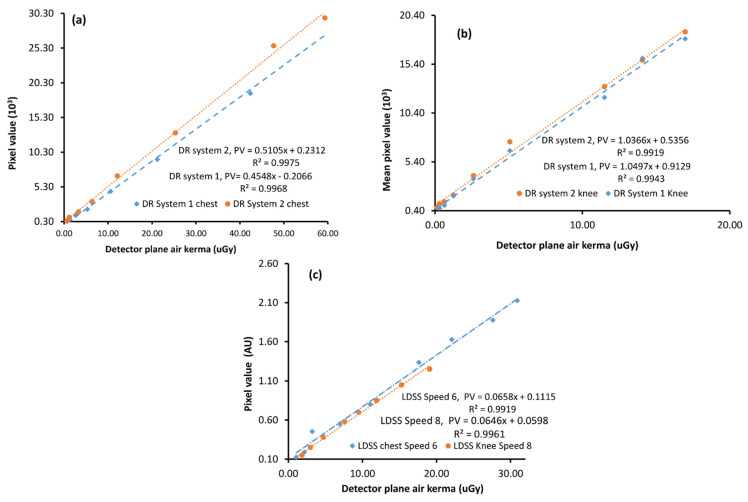
Detector response functions of (**a**) DR imaging systems for the chest protocol, (**b**) DR imaging systems for the knee protocol and (**c**) LDSS X-ray imaging system for both chest (speed 6) and knee (speed 8) protocols, AU = Arbitrary unit.

**Figure 5 diagnostics-11-01699-f005:**
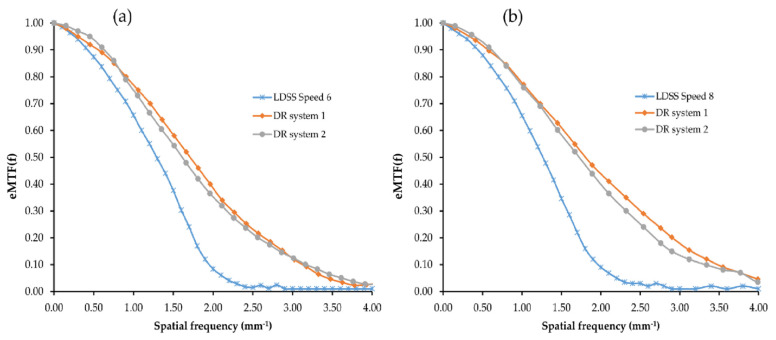
Pre-sampled eMTF comparison for all imaging systems, for the (**a**) chest protocol and (**b**) knee protocol.

**Figure 6 diagnostics-11-01699-f006:**
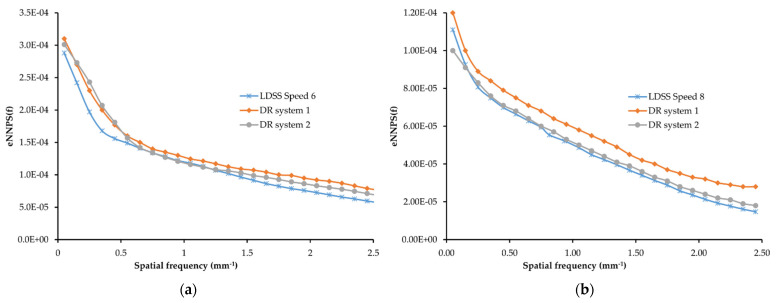
Effective NNPS for all imaging systems: (**a**) for chest and (**b**) for knee protocols.

**Figure 7 diagnostics-11-01699-f007:**
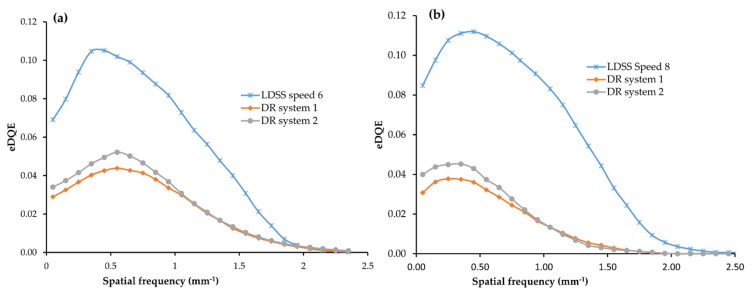
Comparison of eDQE, (**a**) in chest protocol for all imaging systems, (**b**) in knee protocol for all imaging systems.

**Figure 8 diagnostics-11-01699-f008:**
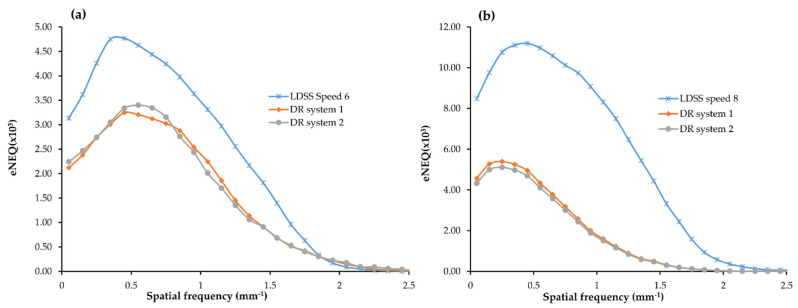
Comparison of eNEQ (**a**) chest protocol for all imaging systems, (**b**) knee protocol for all imaging systems.

**Figure 9 diagnostics-11-01699-f009:**
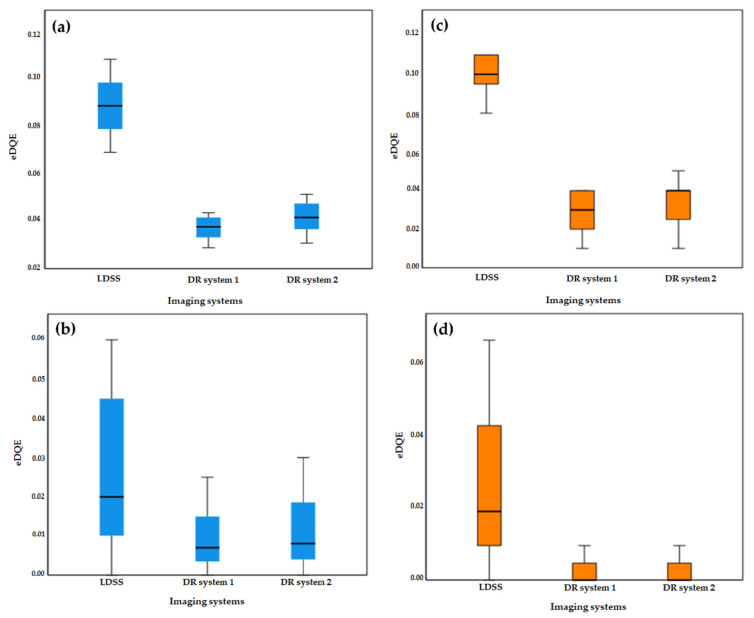
Box-plot of eDQE for (**a**) low frequency in knee protocol, (**b**) high frequency in knee protocol and (**c**) low frequency in knee protocol and (**d**) high frequency in knee protocol.

**Figure 10 diagnostics-11-01699-f010:**
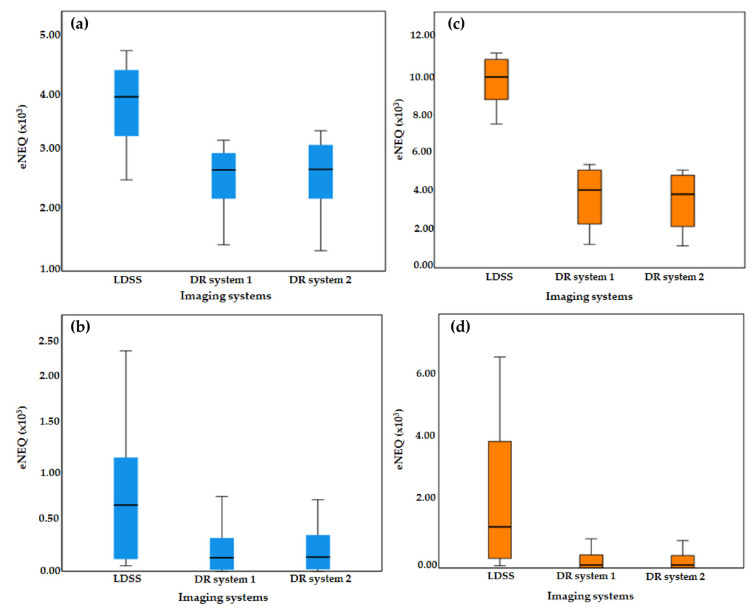
Box-plot of eNEQ values for (**a**) low frequency (0.00–1.25), (**b**) high frequency (1.35–2.55) in the chest protocol and (**c**) low frequency (0.00–1.25), (**d**) high frequency (1.35–2.55) in the knee protocol.

**Table 1 diagnostics-11-01699-t001:** The list of study protocols and radiation dose level settings used in both patient dose and image quality comparison of the LDSS and DR systems. PA = posterior/anterior projection.

Systems	Projections	Dose Level	Protocols
DR system 1	PA	Standard (default)	Chest
DR system 2	PA	Standard (default)	
LDSS	PA	Medium dose (speed 6)	
DR system 1	PA	Standard (default)	Knee
DR system 2	PA	Standard (default)	
LDSS	PA	High dose (speed 8)	

**Table 2 diagnostics-11-01699-t002:** Exposure settings in chest and knee examination protocols for all imaging systems, n/a = not applicable, AEC = automatic exposure control, SID = source to image distance, AF = additional filtration, Al = aluminium filtration and Cu = copper filtration.

Imaging Systems	Tube Voltage [kV]	SID (cm)	Tube Current (mA)	Tube Load (mAs)	Exposure Mode	AF Al/Cu (mm)	Examination Protocols
LDSS speed 6	90	130	280	n/a	Manual	0/0.1	Chest
DR system 1	133	250	n/a	1.6	AEC	1/0.2	
DR system 2	145	300	n/a	1.8	AEC	0/0.2	
LDSS speed 8	68	130	400	n/a	Manual	0/0	Knee
DR system 1	57	110	n/a	8.5	Manual	0/0	
DR system 2	63	115	n/a	6.3	Manual	0/0	

**Table 3 diagnostics-11-01699-t003:** All assessed parameters in relation to the calculation of eDQE in the horizontal plane. Transmission factor (TF), air kerma (AK), source image distance (SID), scatter fraction (SF) and detector air kerma (DAK) (no scatter).

Imaging Systems	SID (cm)	Tube Voltage (kVp)	TF	*q*-Value (mm^−2^ µGy^−1^)	SF	AK (µGy)	DAK (µGy)	Max eNEQ (×10^3^)	Max eDQE	Clinical Protocols
LDSS Speed 6	130	90	0.10	31170	0.05	13.96	1.46	4.77	0.10	Chest
DR system 1	250	141	0.12	37172	0.19	16.40	1.97	3.24	0.05	
DR system 2	300	145	0.11	38093	0.18	15.68	1.71	3.40	0.05	
LDSS Speed 8	130	68	0.14	27520	0.03	15.72	2.25	11.19	0.11	Knee
DR system 1	115	57	0.15	29892	0.28	18.40	2.96	5.10	0.04	
DR system 2	110	63	0.16	30228	0.26	17.90	2.70	5.39	0.05	

**Table 4 diagnostics-11-01699-t004:** Kruskal–Wallis multiple comparison of eDQE values in both low and high frequencies for all imaging systems in both knee and chest protocols.

Protocols	Frequency Range (mm^−1^)	Total Number	Test Statistic	Degree of Freedom	*p*-Value
chest	0.00–1.00	33	22.63	2	**<0.001**
chest	1.15–2.15	33	6.25	2	**0.044**
knee freq	0.00–1.00	33	22.32	2	**<0.001**
knee freq	1.15–2.15	33	16.61	2	**<0.001**

**Table 5 diagnostics-11-01699-t005:** Mann–Whitney U statistical pairwise comparison of eDQE values at both low and high frequencies for the three imaging systems in chest and knee protocols.

Systems Comparison	Test Statistics	Knee Freq.(0–1)	Knee Freq.(1.15–2.15)	Chest Freq.(0–1)	Chest Freq.(1.15–2.15)
LDSS vs. DR system 1	Mann–Whitney U	0.00	11.50	0.00	27.00
	Z	–4.02	–3.37	–3.98	–2.21
	***p*-value**	**<0.001**	**<0.001**	**<0.001**	**0.03**
LDSS vs. DR system 2	Mann–Whitney U	0.00	11.50	0.00	29.00
	Z	–4.03	–3.37	–3.98	–2.08
	***p*-value**	**<0.001**	**< 0.001**	**<0.001**	**0.04**
DR sys. 1 vs. DR sys. 2	Mann–Whitney U	48.50	60.50	35.50	57.00
	Z	–0.83	0.00	–6.65	–230
	***p*-value**	**0.44**	**0.99**	**0.10**	**0.85**

**Table 6 diagnostics-11-01699-t006:** Kruskal–Wallis multiple comparison of eNEQ values at both low and high frequencies for all imaging systems in both the knee and chest clinical protocols.

Protocols	Frequency Range (mm^−1^)	Total Number	Test Statistic	Degree of Freedom	*p*-Value
chest	0.00–1.25	39	16.51	2	**<0.001**
chest	1.35–2.55	39	6.52	2	**0.038**
knee	0.00–1.25	39	23.5	2	**<0.001**
knee	1.35–2.55	39	12.1	2	**0.002**

**Table 7 diagnostics-11-01699-t007:** Mann–Whitney U statistical pairwise comparisons of eNEQ values at low and high frequencies for the three imaging systems in both chest and knee protocols.

Systems Comparison	Test Statistics	Knee Freq.(0.00–1.25)	Knee Freq.(1.35–2.55)	Chest Freq.(0.00–1.25)	Chest Freq.(1.35–2.55)
LDSS vs. DR system 1	Mann–Whitney U	0.00	20.50	14.00	40.00
	Z	–4.16	–3.00	–3.62	–2.28
	***p*-value**	**<0.001**	**0.002**	**<0.001**	**0.022**
LDSS vs. DR system 2	Mann–Whitney U	0.00	20.50	19.00	43.50
	Z	–4.16	–3.00	–3.36	–2.10
	***p*-value**	**<0.001**	**0.002**	**<0.001**	**0.034**
DR system 1 vs. DR system 2	Mann–Whitney U	62.00	69.00	78.50	81.50
	Z	–0.58	–0.18	–0.31	–0.15
	***p*-value**	**0.59**	**0.89**	**0.76**	**0.80**

## Data Availability

Results and data of this study were not provided in any public places. The phantom images, calculations and reports were archived in a personal computer hard-drive.

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
