# Peer review of "Quantitative Image Quality Metrics of the Low-Dose 2D/3D Slot Scanner Compared to Two Conventional Digital Radiography X-ray Imaging Systems"

_diagnostics, 2021, doi:10.3390/diagnostics11091699_

Round 1

Reviewer 1 Report

In this manuscript, the authors compared performances of the LDSS imaging system and two conventional DR imaging systems by measuring eDQE and eNEQ. The manuscript is well organized and clearly written to efficiently provide comparison results. For improving the quality of the manuscript, several comments are provided as follows.

  1. A brief comparison of the system complexities is required. In other words, differences in the detector structure, x-ray tube and its scanning mechanism, and overall implementation cost.
  2. What are the pros and cons of the LDSS system compared to the conventional DR systems besides eDQE and eNEQ?
  3. The eMTF performance of LDSS is usually lower than the conventional DR case. On the other hand, the noise property of LDSS is much better. A brief discussion on these observations will be useful.

Several minor comments as follows.

  1. On Line 52, what is reason for ‘not yet generally used’?
  2. In Table 2, what is the difference between LDSS speed 6 and speed 8?
  3. In (1), is there any special meaning in using f’ instead of f?
  4. In (4) and (5), using the same symbol k could be confusing.
  5. In (9), phi should be Phi.

Author Response

Dear reviewer 

Thank you very much for your valuable comments and suggested corrections. Please find the attached document contain the response to your comments

Reviewer 2 Report

Dear authors, 

I find your paper interesting and well written. Nonetheless, I have several minor comments to the matter and to the language (bottom). Hopes they will help to improve the clarity of your paper.

Yours sincerely.

Comments to the matter:

In Table 2, it would be nice if you could unify the information about the current, i.e., indicate the same parameter (tube current or tube load) for all systems used, as in the present form, the parameters cannot be really compared (280 mA @ LDSS vs. 1.6 mAs @ DR1).

Page 5, line 169: you say that the images were acquired with a range of 10 different dose levels. What did change? The current/time, or even the tube voltage? Please, specify.

Figure 3: If the phantom used are the same for a) and b), c) and d), please, provide the appropriate dimensions in the image. The knee phantom seems to be completely different in c and d.

Section 3.1: you talk about the response of the detector, but you do not mention that you are talking about the response to the dose (or air kerma). It is later clear from the figures, but I would specify this anyways (since the detector response to the intensity, tube voltage, tube current etc. are also often of interest). Moreover, I do not think this part is correct, as you say that the LDSS data are linearized using a 5th order calibration polynomial identified for the LDSS system. You could probably identify an appropriate polynomial for the DR systems as well, reaching thus better linearity of them. You also say that the LDSS is linearized by slightly data manipulation. Although you provide the reference, I would really welcome a more detailed comment on this topic.

Section 3.3, page 10, line 346: you say that there are less photons reaching the DR detector than the LDSS detectors. Why? Is it because of source-to-detector distance? If it is the case, would it not be better to try to make a setup that would allow to compare the both systems in as-far-as-possible identical conditions?

Section 3.6, page 12, line 387: you reference Figure 7, but should you not reference Figure 9?

Comments to the language:

Page 2, line 67: for scanning systems intead of "for a scanning systems"

Page 2, line 71: imaging system instead of "imagining system"

Page 3, line 124: The NEXT phantom was designed... - please, reformulate this sentence, there are two verbs (was, consists)

Page 3, line 132: To assess the eMTF... - please, reformulate, two verbs (was, meets)

Page 4, line 148 - two times "are"

Page 10, line 332: is there "faster" missing between "acquisition" and "than"?

Page 10, line 345: space between "by" and "higher"; add "which" or "that" between "systems" and "occur"

Page 15, line 473: "for this radiology department" - is there a better word than "department"? Perhaps "branch"?

Page 16, line 506: ThoraScan instead of "TharaScan"

Page 16, line 520: eDQE instead of "eQDE"

Reference 6: The EOS??? imaging system... - please, correct

Author Response

Dear reviewer 

Thank you very much for your valuable comments and suggested corrections. Please find the attached document which contain the response to your comments.
